# A Quantum Annealing Instance Selection Approach for Efficient and Effective Transformer Fine-Tuning

## ABSTRACT

Deep Learning approaches have become pervasive in recent years. In fact, they allow for solving tasks that were thought to be too complex a few decades ago, sometimes with superhuman effectiveness. However, these models require huge datasets to be properly trained and to provide a good generalization. This translates into high training and fine-tuning time, even several days for the most complex models and large datasets. In this work, we present a novel quantum *Instance Selection (IS)* approach that allows to significantly reduce the size of the training datasets (by up to 28%) while maintaining the model's effectiveness, thus promoting (training) speedups and scalability. Our solution is innovative in the sense that it exploits a different computing paradigm – *Quantum Annealing (QA)* – a specific Quantum Computing paradigm that can be used to tackle practical optimization problems. To the best of our knowledge, there have been no prior attempts to tackle the IS problem using QA. Furthermore, we propose a new *Quadratic Unconstrained Binary Optimization (QUBO)* formulation specific for the IS problem, which is a contribution in itself. Through an extensive set of experiments with several *Automatic Text Classification (ATC)* benchmarks, we empirically demonstrate both the feasibility of our quantum solution and its competitiveness with the current state-of-the-art IS solutions.

## CCS CONCEPTS

• **Computer systems organization** → **Quantum computing**; **Quantum computing**; • **Computing methodologies** → **Supervised learning by classification**; *Machine learning approaches*; **Supervised learning by classification**; *Machine learning approaches*; • **Information systems** → *Information retrieval*; *Information retrieval*.

## KEYWORDS

Instance Selection, Quantum Computing, Quantum Annealing, Deep Learning, Text Classification, Transformers

**ACM Reference Format:**
Anonymous Author(s). 2024. A Quantum Annealing Instance Selection Approach for Efficient and Effective Transformer Fine-Tuning. In *Proceedings of ACM SIGIR International Conference on the Theory of Information Retrieval (ICTIR'24).* ACM, New York, NY, USA, 10 pages. https://doi.org/XXXXXXX.XXXXXXX

## 1 INTRODUCTION

Deep Neural Networks and, in particular, Transformers are currently applied in tasks such as Ad-Hoc Retrieval [28] to rank relevant documents [29], *Automatic Text Classification (ATC)* [10] to assign semantic labels to a piece of text, and Sentiment Analysis [3] to detect whether a sentence has a positive, negative or neutral sentiment towards some subject (e.g., a product, movie or restaurant) [27]. Transformers such as BERT, RoBERTa, BART, and others [12, 23, 26] are designed to capture not only the meaning of the words but also their context in a sentence, thus capturing relationships and dependencies between words in a piece of text.

Transformers perform exceptionally well in ATC, Sentiment Analysis, Question Answering [31], and other text-based tasks [10, 32]. To achieve such performance, Transformers and other Large Language Models (e.g., GPT, LLama) [41] rely on very complex pretrained models with several millions of parameters and, although they can also be used in a zero-shot manner, their fine-tuning on a given domain or task is essential to guarantee effectiveness [11]. However, even fine-tuning these models still requires a considerable investment of time and computational resources, especially on *large* datasets. Ameliorating such negative aspects may be achieved by two (not mutually exclusive) different approaches:

- Model Compression or Pruning techniques [25] applied to reduce the complexity of the *Deep Learning (DL)* models, thus reducing the training time while trying to keep the model's effectiveness;
- Instance Selection (IS) techniques [10], used to significantly reduce the training dataset size to speed up the training phase while trying to keep the models' effectiveness unchanged.

Note that both Model Compression and IS are computationally challenging problems, often requiring heuristics and greedy approaches to be solved.

In this work, we focus on the second alternative by proposing a novel IS approach called *Balanced Cosine (BCos)* that leverages *Quantum Annealing (QA)*, a specialized form of *Quantum Computing (QC)*. An innovative aspect of our proposal is to investigate the feasibility and performance of QA to perform IS. Indeed, QC promises to deliver substantial performance benefits with respect to traditional approaches [30] and, QA in particular, is especially suitable for solving computationally intensive problems, provided that they can be formulated as an optimization problem of the *Quadratic Unconstrained Binary Optimization (QUBO)* family. We highlight that QUBO represents a family of problems with given characteristics and the main challenge is to understand how to formulate a general optimization problem as a QUBO problem. To the best of our knowledge, there has not been other previous work aiming at solving IS by using QA technologies, and BCos represents the first formulation of IS as a QUBO problem, suitable for QA.

In this context, we run a comprehensive series of experiments on several ATC benchmarks with different characteristics (e.g., size, number of classes, class balancing) to demonstrate the feasibility

and applicability of our proposed quantum IS approach. We also compare our solutions with the *State-Of-The-Art (SOTA)* approaches in the IS field. Our experimental results reveal that BCos is very competitive, being able to reduce the training sets by up to 28%, achieving speedups of up to 1.35x (which can be further improved as QC technology advances), while keeping the effectiveness of the trained model in most cases. Note that our approach has been designed and experimented with in the context of ATC tasks, but its underlying principles can be useful for other *Information Retrieval (IR)* or *Natural Language Processing (NLP)* tasks where Transformer-based solutions are used, including modern LLMs.

In sum, the main contributions of this work are:

- A proposal of a novel IS approach that employs QA, demonstrating the feasibility of our solution in a very different and emerging computing paradigm, thus serving as a proof-of-concept about how complex but practical problems can be solved with such quantum technologies;
- The first QUBO formulation of the IS problem;
- An extensive experimentation and comparative performance study under several criteria (effectiveness, efficiency, scalability) of our quantum approach against SOTA approaches in the field.

It is important to stress that a critical aspect of quantum technologies is that they are still in their early stages of development [17] and they still suffer from several limitations when it comes to hardware capabilities, e.g., limitations in the number of qubits and their topology, sensitivity to external noise and decoherence, errors and stochasticity of the results [1, 30]. Therefore, the challenge is not only to find a way to formulate an algorithm such that it can be computed using quantum technologies but also a way which accounts for and works around the current limitations of the hardware which, in turn, affects the overall performance achieved. On the other hand, traditional hardware has been studied and improved for several decades, leading to very robust and consistent performance. Therefore, while computational supremacy for QA begins to be observed and it is expected to fully happen in the next few years [19, 20], it is still not possible to observe substantial performance gains under all circumstances. As a consequence, improvements observed for QC technologies should be considered just as the tip of the iceberg rather than the maximum that can be achieved as this technology continuously progresses and evolves. In other words, demonstrating early practical and theoretical developments and results, even with current limitations, is key in foreseeing the potential ahead.

This work is organized as follows. Section 2 provides an overview of Quantum and Simulated Annealing and includes IS related work. Section 3 details our approach. Section 4 describes the experimental setup. Section 5 presents and discusses the achieved results. Section 6 draws conclusions and discusses potential future work.

## 2 BACKGROUND AND RELATED WORKS

We provide a brief introduction to QA. We also explain *Simulated Annealing (SA)*, a traditional optimization algorithm that does not take advantage of quantum technologies. Finally, we discuss some related work for IS.

### 2.1 Quantum Annealing

QA is a QC paradigm that is based on special-purpose devices (quantum annealers) able to tackle optimization problems with a certain structure, such as the famous Travelling Salesman Problem. The basic idea of a quantum annealer is to represent a problem as the energy of a physical system and then leverage quantum-mechanical phenomena, e.g., superposition and entanglement, to let the system find a state of minimal energy, which corresponds to the solution of the original problem.

To use quantum annealers, one needs to formulate the optimization problem as a minimization one using the *Quadratic Unconstrained Binary Optimization (QUBO)* formulation [14], a well-known optimization technique. QUBO is defined as:

$$\min \quad y = x^T Q x \tag{1}$$

where $x$ is a vector of binary decision variables, and $Q$ is a matrix of constant values representing the problem we wish to solve. We emphasize that formulating an optimization problem (e.g., the IS problem) as a QUBO problem is not trivial. Once the problem has been formulated as QUBO, a further step called *minor embedding* is required to map the general mathematical formulation into the physical quantum annealer hardware, accounting for the limited number of qubits and the physical connections between them. Each quantum annealer has, in fact, its own architecture, which can be seen as a graph: each vertex represents a qubit, and each edge represents an interaction between 2 qubits. Therefore, we need to proceed with the *minor embedding* phase that consists of adapting the problem $Q$ to the physical architecture we have at our disposal. This involves choosing which qubits represent our variables as well as duplicating qubits if the number of physical connections is lower than the number of interactions between the variables in $Q$. Minor embedding is a complex task in itself and a *NP*-hard problem, which can be solved relying on some heuristic methods [7].

To sum up, using a quantum annealer requires several stages [44]:

**Formulation:** find a way to express the desired algorithm as an optimization problem by leveraging the QUBO framework;

**Computation:** compute the actual QUBO matrix $Q$ needed to solve the optimization problem (our algorithm);

**Embedding:** generate the minor embedding of the QUBO for the quantum annealer hardware;

**Data Transfer:** transfer the problem and the embedding on the global network to the data center that hosts the quantum annealer;

**Annealing:** run the quantum annealer itself. This is an inherently stochastic process. Therefore, it is usually run a large number of times (hundreds) in which several samples are returned, each one resembling a possible solution to the considered problem. The solutions must then be checked for their feasibility, and then the best one among them (i.e., the optimal one according to the objective function) is usually considered as the final solution to the submitted problem.

Generally, a QUBO problem can be solved by a quantum annealer in a few milliseconds.

Occasionally, it might be necessary to add further constraints to the problems. This can be done by means of penalties P($x$) [44], which penalize solutions that do not meet the specified constraints.

These penalties are then added to the original cost function $y$ to achieve the final formulation as follows:

$$\min \quad C(x) = y + P(x) . \tag{2}$$

Penalties can be controlled through hyperparameters to manage their influence with respect to the given formulation. A general rule is that they should be *high enough* to be effective but should not be *too high* in order to avoid introducing distortions and noise.

*Applications.* QA has already been employed in IR, RecSys, and NLP tasks such as Feature Selection [13, 33] and Clustering [21], showcasing potential benefits in terms of efficiency compared to traditional hardware alternatives. CLEF is also running a new lab on feature selection and clustering using QA [34, 35]. To the best of our knowledge, no previous work has investigated QA for IS yet.

## 2.2 Simulated Annealing

SA is a consolidated algorithm that can be run on traditional hardware [6, 42]. It is a probabilistic algorithm that can be used to find the global minimum of a given cost function, even in the presence of many local minima. It is based on an iterative process that starts from an initial solution and tries to improve it by randomly perturbing it. The cost function is represented by the QUBO problem formulation, similar to what would be used for QA. It takes inspiration from annealing in metallurgy, a technique that consists of heating and slowly lowering the temperature of a material to alter its physical properties. This also translates into minimizing the system's energy. In SA, there is no minor embedding phase since the problem is directly solved on a traditional machine.

We underline that SA is an optimization algorithm different from QA, it is not a simulation of QA on a traditional machine, and, therefore these two algorithms are not equivalent. However, SA can be used for benchmarking purposes to show how well QA performs with respect to a traditional hardware counterpart.

Access to quantum annealers is limited to ensure a fair distribution of resources. Therefore, SA can also be used to perform initial experiments to assess a QUBO formulation feasibility without affecting the available quota in the quantum environment.

## 2.3 Instance Selection

In this section, we briefly describe some IS SOTA approaches. IS has received attention lately due to the increasing computational (and environmental) costs of training and tuning large language models, which are useful for many IR and NLP tasks.

In [8], the authors compared the most traditional and recent IS approaches, applying them in the ATC context, with a special focus on the impact of IS on transformers. The analysis focused on the capability of those approaches to find a good balance among three factors: effectiveness, training size reduction, and efficiency (speed up) (the "tripod" constraints). For our work, we selected as baselines the four best methods found in [8] based on this tradeoff: **E2SC**, **CNN**, **LSSm**, and **LSBo**.

The Effective, Efficient, and Scalable Confidence-Based Instance Selection Framework **(E2SC)** [8] relies on efficient and calibrated weak classifiers to remove *redundancy* from the training set. The authors correlate redundancy with the confidence of a weak and calibrated classifier in determining the class of a training instance:

the more confident, the higher the likelihood that other instances in the training have similar information to the instance to be removed. E2SC was the method that obtained the best results in terms of the tripod requirements in a recent comparison[8], being currently considered the state of the art in the field. The time complexity of E2SC is $O(log(n))$, where $n$ is the size of the original set, being one of the most scalable solutions among our baselines.

The Condensed Nearest Neighbor (**CNN**) [16], a traditional IS technique, focuses on harder instances, defined as those misclassified by an internal classification process. The intuition behind CNN is that instances near the classification boundaries are more representative for the sake of training a classifier. The time complexity of CNN is $O(n^3)$. In this paper, we focus on similar ideas but avoid the cost of internally running a classifier multiple times using a different optimization formulation.

In [24], the authors proposed Local Set-based Smoother (**LSSm**), and Local Set Border Selector (**LSBo**). Both methods are based on the idea of Local Sets (LS), which consists of instances within a sub-region of the feature space hyperplane that belong to the same class. In LSSm, the authors derive the LS concept into the usefulness $(u)$ and harmfulness $(h)$ of each instance. LSSm aims at keeping instances with higher levels of importance and influence on others $(u > h)$. The time complexity of LSSm is $O(n^2)$. LSBo first performs noise removal by applying LSSm, and then it sorts the remaining instances according to the LS Cardinality (LSC). Consequently, instances within the classification boundaries will be inserted in the final solution since they have lower LSC. Similar to LSSm, LSBo's time complexity is $O(n^2)$. We use both methods as baselines.

## 3 METHODOLOGY

Our BCos approach is based on the following ideas:

- Given a training set $T$ with size $|T|$, we aim at reducing its size by a factor $p \in ]0, 1[$ such that the size of the reduced subset $t$ produced using our approach is $|t| = p \times |T|$;
- The subset $t$ must represent well the original full training dataset $T$. Ideally, it should contain *representative samples* of the original set so that the Transformer can be trained on $t$ and learn a similar set of patterns as it would do with the full training set $T$ but at a reduced computational cost. There are several possible strategies to produce such reduced set $t$, such as removing redundant samples [8], a variant which we exploit here, complemented with heuristics to find *difficult instances*, defined based on sets of pairs of similar documents belonging to distinct classes. These documents are likely outliers or lie in the classes' boundaries.

## 3.1 The QUBO formulation

As pointed out in Section 2.1, to solve a problem with a quantum annealer, we first need to transform it into an optimization problem and then to its corresponding QUBO formulation. Note that QUBO is a general framework in optimization and formulating a problem as QUBO is neither always possible nor immediate or done automatically; therefore, the QUBO formulation of a problem is an innovative contribution in itself.

Our QUBO formulation follows the general framework in Equation 1, where $x \in \{0, 1\}$ represents whether a document should be

removed (0) or kept (1), and the matrix $Q$ is defined as follows:

$$Q_{i,j} = \begin{cases} \cos(doc_i, doc_j) & \text{if } lbl_i = lbl_j \text{ and } i \neq j \\ -\cos(doc_i, doc_j) & \text{if } lbl_i \neq lbl_j \text{ and } i \neq j \\ \frac{|T[lbl_i]|}{|T|} & \text{if } i = j \end{cases} \quad (3)$$

where $doc_i$ is the vectorized version of the $i$-th document in the training set $T$, $lbl_i$ is the label (or class) associated with the $i$-th document in the training set, $|T[lbl_i]|$ is the number of documents in the training set having the same label as the $i$-th document in the training set. Furthermore, $\cos(doc_i, doc_j)$ represents the cosine similarity between two vectorized documents. Each element in Equation 3 has its own intuitive interpretation, remembering that we are solving a minimization problem:

(1) The more similar two documents belonging to the same class are, the more likely they will be removed from $T$. Therefore, we assign them a **positive** value in the QUBO matrix corresponding to their cosine similarity;

(2) The more similar two documents belonging to different classes are, the more likely we need to keep them in $T$. Therefore, we assign them a **negative** value in the QUBO matrix corresponding to their cosine similarity. This is done since there is a high probability that these two documents represent difficult instances, lying close to the classification boundaries[1];

(3) In the cases that $T$ has a skewed class distribution, we want to avoid removing documents from minority classes, which could worsen the imbalance in $T$. Imbalanced distributions are known to be a big issue in classification tasks due to the bias towards the majority classes [15]. Therefore, in our formulation, we try to penalize the removal of documents belonging to minority classes with respect to others.

## 3.2 Control the number of selected instances

To complete the QUBO formulation (Equation 1), we need to add the constraint of selecting a subset of a predefined size as explained in Section 3. In this case, we need to set opportune constraints that allow us to keep only $\lfloor p \cdot |T| \rfloor$ documents out of $|T|$ documents. This can be achieved as follows:

$$\gamma \cdot \left( \sum_i a_i \cdot x_i - \lfloor p \cdot |T| \rfloor \right)^2 = 0 , \quad (4)$$

with $\gamma$ representing a *penalty* sufficiently high. This translates into summing these constraints to the original QUBO formulation.

In our case, the percentage of documents that are kept is 75% of the total collection since a reduction of 20-25% has been shown to be a good trade-off between efficiency and effectiveness [8]. We leave for future work to study the tradeoffs of effectiveness vs. speedup of other more aggressive reduction rates.

## 3.3 Batches

As anticipated in Section 1, QC still suffers from several hardware limitations. In our case, we employ quantum annealers provided by D-Wave, one of the leading companies in the sector [39], whose most performing quantum annealer is the D-Wave Advantage, with about 5,000 qubits. On the other hand, the size of a training set is

[1]The other possibility is that one of the documents is an outlier, a probability that we neglect for now, leaving for future work to deal with it

typically in the order of thousands to tens of thousands of documents, which cannot fit in the current hardware and thus requires to figure out a way to proceed in batches.

Furthermore, as explained in Section 2.1, the QUBO matrix $Q$ needs to be mapped onto the physical layout of the qubits in the quantum annealer via the minor embedding process and this significantly lowers the number of variables which can be actually processed in a batch. In particular, in the case of D-Wave, the 5,000 qubits allow the representation of approximately a hundred variables at a time, depending on the actual layout of $Q$. Therefore, we decided to set a batch size of $B = 80$ document instances, in order to use most of the capacity of the quantum annealer but, at the same time, not risk exceeding it for some unfortunate layout of $Q$. We then split the overall problem into $n = \lceil |T|/B \rceil$ batches; note that the last $n$-th batch might contain less than $B$ document instances and, in particular, equal to $|T| \mod B$.

For each batch, we extract a subset $sub_i$ of the most relevant documents and, since there is no intersection among the batches, we consider their union as the final subset of documents that should be fed to the Transformer during the training phase:

$$S = \bigcup_{i=1}^{n} sub_i . \quad (5)$$

According to the workflow explained in Section 2.1, each batch $B$ would need to undergo the minor embedding process, which, as explained, is a computationally demanding task in itself. Since we have to process in the order of hundreds of batches, repeating the minor embedding process over and over would undermine the overall performance of QA. However, the minor embedding process is concerned with the topology of the graph represented by $Q$, e.g., which variables are connected within $Q$, with respect to the topology of the physical qubits, e.g., how they are connected at the hardware level, rather than with the actual values contained in $Q$.

In our case, the $Q$ matrix of each batch $B$ is a fully connected graph, whose structure is the same for all the batches. The only exception might be the $n$-th batch if its size is less than $B$, since its $Q$ matrix would represent a smaller graph, but always fully connected.

Therefore, we can avoid repeating the minor embedding process over and over and perform it at most two times: one time for any batch of size $B$ and once more for the last $n$-th batch of size less than $B$, if needed. In this way, we can avoid many useless computations and save a substantial amount of processing time for QA.

Finally, note that this batching approach is also beneficial for the traditional computation case. Indeed, we can use the same batches and $Q$ matrices also for running the SA optimization algorithm, which, in this way, will be able to benefit from the parallelism provided by the multiple CPUs available on the traditional hardware on which it is executed.

## 3.4 Visualization Example of BCos

To better understand how BCos works and to visualize which instances BCos removes, we created a synthetic dataset of 2,000 two-dimensional documents belonging to two balanced classes. Figure 1a shows the generated instances while Figure 1b shows how the removed points are, in most cases, "redundant". Indeed, we can consider the contribution of these points to the classifier learning as

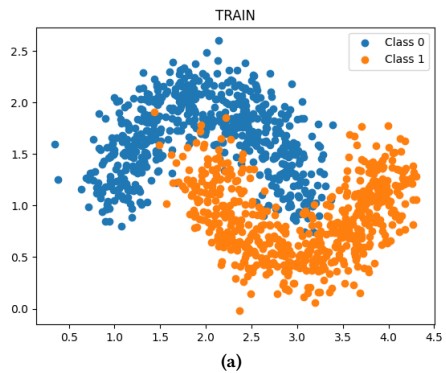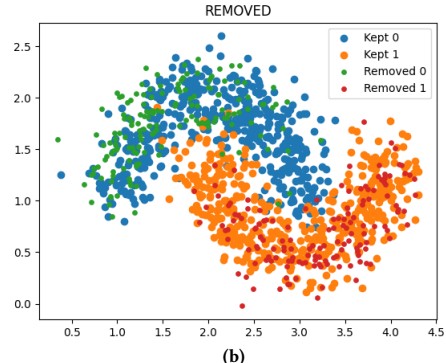

**Figure 1: The dataset composed of 2D points (a) and the visualization of the points that are kept and removed by BCos (b)**

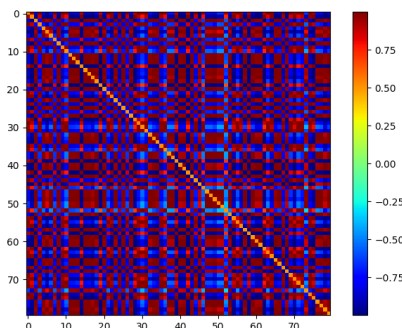

**Figure 2: A visualization of the resulting QUBO matrix extracted from a batch of 80 documents.**

**Table 1: Datasets characteristics.**

| Considered datasets and their characteristics. | | | |
|---|---|---|---|
| Dataset | Size | N° of Labels | Class Skewness |
| Vader NYT | 4946 | 2 | Almost balanced |
| Yelp Reviews | 5000 | 2 | Balanced |
| WebKB | 8199 | 7 | Imbalanced |
| OHSUMED | 18302 | 23 | Imbalanced |
| 20 Newsgroups | 18846 | 20 | Balanced |
| AG News | 127600 | 4 | Balanced |

### 4.1 Datasets

Our approach considers different datasets, which are reported in detail in Table 1. As it is possible to see, each dataset has its own characteristics, which allows us to measure the performance of our approach in different scenarios. Therefore, we can compare it with other approaches in a fair way, ensuring the reliability and applicability of our results also in the case of other datasets.

### 4.2 Data Representation and Preprocessing

We process each dataset to obtain a configuration that is suitable for our needs. This process is described in Figure 3.

*Dataset splits.* To obtain scientifically sound results and allow for their statistical analysis, we adopt a 5-fold validation experimental setup. In this way, we obtain 5 different Training sets and 5 Test sets. From each of the 5 training sets, we extracted a validation set (10% of the size of the corresponding Training set) that is employed to avoid overfitting when fine-tuning the BERT model.

*Data Representation.* In BCos, we convert the training datasets consisting of textual documents into datasets of embeddings generated using a pre-trained BERT model according to the Zero-shot learning paradigm. Each document/piece of text is, therefore, converted into a numerical vector of 768 dimensions. In the other approaches, a TF-IDF [2] representation is employed to convert documents into numerical vectors, making use of *scikit-learn* [36] – we removed stopwords and kept features appearing in at least two documents and, then e normalized the TF-IDF product result using the L2-norm. In fact, regarding the baselines, TF-IDF representation has been shown to be more efficient and effective with respect to contextual embeddings [10]. More specifically, the authors demonstrated

already provided by other points of the dataset, making them sort of "easy" points not lying on the frontier between the two classes but instead in the "middle" or center of the classes.

Figure 2 shows the corresponding QUBO matrix for one batch of $B = 80$ instances. Each element $Q_{i,j}$ represents the relation between $doc_i$ and $doc_j$ according to equation 3. We can observe that it is a symmetric matrix, being $Q$ fully connected. Moreover, since the two classes are balanced, we can observe how the elements on the diagonal have (almost) the same value.

## 4 EXPERIMENTAL SETUP

The experiments were executed on an AMD Ryzen 5 5600X Processor with 6-Core and 12-Threads, running at 3.70GHz, 64Gb RAM, and a NVIDIA RTX 3090 24 GB. The quantum annealer employed for the experiments is the D-Wave Advantage ($\approx 5,000$ qubits).

To promote reproducibility, all code, documentation, and datasets will be available on GitHub[2] in case of acceptance. However, the API key we have used to access D-Wave's quantum annealers cannot be disclosed. Therefore, to employ quantum annealers in our code, an API key must be requested from D-Wave.

---

[2]https://github.com/<double-blind>

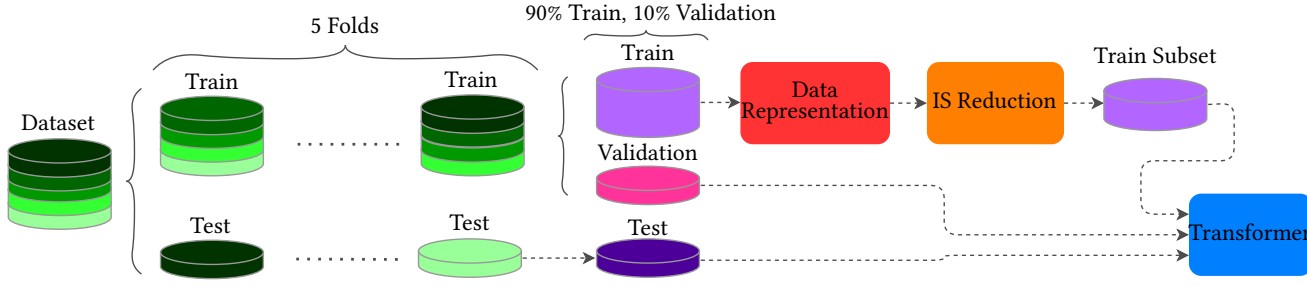

**Figure 3: The overall process, starting from the division of the dataset into 5 folds to the final BERT Fine-tuning and Testing.**

that using contextual embeddings: (a) led the baselines to become 1.3x to 3.0x more expensive than training the model with all data; and (b) caused statistically significant losses in more than half of the tested datasets. Nevertheless, in the case of BCos, preliminary experiments empirically have shown that the use of embeddings led to better effectiveness but increased overall IS time. To summarize, we adopted the best representation (TFIDF or BERT-based embeddings) as input for each in individual IS approach.

### 4.3 Transformer-based Text Classification

BERT [12] is a comprehensive DL classifier that follows an end-to-end architecture. It features a bidirectional Transformer encoder with 24 Transformer blocks, 1024 hidden layers, and a total of 340 million parameters. The model undergoes pre-training on a massive 3.3 billion-word corpus. BERT employs a multi-layer bidirectional Transformer encoder, where the self-attention layer functions both forward and backward. At the time of its launch, BERT redefined the state of the art across 11 NLP tasks, and for this reason, we adopted it as our Transformer-based text classification model. We intend to test with other Transformers such as RoBERTa [26], BART [23], XLNet [43] in the near future, but recent benchmarks [10, 11] have shown that the differences among the latest version of these Transformers in some the datasets we use in our experiments are very small (between 1-2p.p.) at a higher cost to fine-tune. Indeed, some benchmarks such as GLUE (https://gluebenchmark.com/) and other recent studies[5] do not make clear even if recent LLMs are better than 2nd generation Transformers such as RoBERTa in tasks such as Sentiment Classification, which we exploit in our experiments.

Due to the considerable number of hyperparameters requiring tuning, conducting a grid search with cross-validation becomes impractical. Consequently, in order to identify the optimal hyperparameter, we adopted the approach outlined in [9]. The best hyperparameters were defined as learning rate to $5 \times 10^{-5}$, the maximum number of epochs to 20, patience to 5 epochs, max length to 256, and batch size to 32.

### 4.4 Instance Selection Baselines

We consider as baselines a set of **4** SOTA IS baseline methods described in Section 2.3, namely: *Condensed Nearest Neighbor* (**CNN**); *Local Set-based Smoother* (**LSSm**); *Local Set Border Selector* (**LSBo**); and the Effective, Efficient, and Scalable Confidence-Based Instance Selection Framework (**E2SC**). All hyperparameters for the IS methods (when applicable) were defined with grid-search, using cross-validation in the training set, as suggested in [8].

### 4.5 Measures and Experimental Protocol

Our objective is to evaluate IS methods based on their ability to reduce both the training set and training (fine-tuning) time while keeping the effectiveness of the Transformer model.

To evaluate classification effectiveness, we adopt Macro Averaged F1 (MacroF1)[40] due to the datasets' skewness. We employ the paired t-test with 95% confidence level to compare average outcomes in our cross-validation experiments. Finally, to adjust for multiple tests, we employed the Bonferroni correction [18].

To evaluate the cost-effectiveness tradeoff, we also consider the total time[3] required to build each model. The speed-up $S$ is calculated as $S = \frac{T_{wo}}{T_w}$, where $T_w$ is the total time spent on model construction using the IS approach, and $T_{wo}$ is the total time spent on execution without the IS phase. It is important to notice that the IS phase time also comprises the time for building the input representation (TF-IDF or BERT-embeddings). Since building BERT-based embeddings is more costly than TF-IDF, as we shall see, the speedups attained with BCos are showcased, both with and without considering the initial BERT embeddings extraction phase. Extracting the embeddings is a complex process that impacts the overall execution time but allows BCos to achieve improved effectiveness.

## 5 EXPERIMENTAL RESULTS

We discuss the results achieved by QA and SA. In particular, we apply our BCos approach to the datasets presented in section 4.1, and compare the results with those produced by current SOTA approaches. We further discuss whether our quantum solution can actually provide benefits in terms of efficiency and effectiveness.

### 5.1 Reduction Rate

Table 2 reports the reduction rates for different methods;

**Table 2: Percentage of reduction of the training set size.**

| Percentages of reduction of the IS approaches. | | | | | |
|---|---|---|---|---|---|
| Dataset | E2SC | LSBo | LSSm | CNN | BCos (SA) | BCos (QA) |
| Vader NYT | 25.0% | 48.2% | 6.3% | 32.7% | 25.3% | 27.7% |
| Yelp Revi. | 25.0% | 69.3% | 19.7% | 45.3% | 25.0% | 27.8% |
| WebKB | 25.0% | 70.9% | 24.0% | 42.9% | 25.1% | 28.1% |
| 20 NewsG. | 25.0% | 23.4% | 0.5% | 27.9% | 25.1% | 27.8% |
| OHSUMED | 25.0% | 69.4% | 22.3% | 44.9% | 25.0% | 28.3% |
| AG News | 25.0% | - | - | - | 25.0% | 28.1% |

---

[3]We define total time as the time of IS (when applied) + Transformer fine-tuning.

**Table 3: Macro-F1 scores achieved by BERT on different datasets processed according to SOTA IS approaches. Elements marked with ● indicate the considered model is statistically equivalent to the model trained on the original dataset.**

| BERT f1-macro scores on different datasets processed with different IS approaches. | | | | | | | |
|---|---|---|---|---|---|---|---|
| Dataset | Original Dataset | E2SC | LSBo | LSSm | CNN | BCos (SA) | BCos (QA) |
| Vader NYT | 80.9(1.1) | 81.0(2.0) ● | 79.9(2.4) ● | 80.5(3.3) ● | 79.4(3.1) ● | 80.9(0.8) ● | 81.7(1.9) ● |
| Yelp Reviews | 95.8(0.3) | 96.0(0.7) ● | 94.5(0.6) ● | 94.7(0.9) ● | 94.2(1.4) ● | 95.7(0.7) ● | 95.5(0.6) ● |
| WebKB | 83.3(2.8) | 82.5(2.5) ● | 77.1(1.4) ▼ | 80.1(3.8) ● | 80.2(1.8) ● | 82.6(1.8) ● | 82.5(3.0) ● |
| 20 Newsgroups | 82.3(1.1) | 80.4(0.7) ● | 80.9(0.5) ● | 82.2(0.7) ● | 77.4(1.6) ▼ | 80.6(0.5) ● | 80.6(0.6) ● |
| OHSUMED | 75.5(0.9) | 74.4(0.7) ● | 67.3(2.2) ▼ | 72.2(1.5) ▼ | 69.8(2.1) ▼ | 74.2(0.6) ● | 73.5(1.6) ● |
| AG News | 91.7(0.2) | 91.5(0.2) ● | - | - | - | 91.4(0.3) ● | 91.4(0.2) ● |

As stated, for those methods with pre-fixed reduction rates (E2SC and BCos), we decided to apply a fixed reduction rate of 25% in order to not impact the effectiveness of the Transformer while trying to provide a consistent speed-up, as suggested in [8]. However, it is important to stress that quantum annealers do not ensure to perfectly meet the reduction rate specified. This is due to noise, randomness, and other external factors that impact quantum computation. Indeed, in most cases, we obtained a bit higher reduction rate than the one specified at the beginning, around 28%; as a consequence, the effectiveness of the QA might be slightly lower than what it would have been with an exact 25%.

Each one of the remaining baselines (CNN, LSSm, LSBo) has its own automatic reduction criteria. In this sense, LSBo is the method that provides the higher average reduction rate (56.4% - varying between 23.4% and 70.9%). However, as we shall see, this over-reduction leads to effectiveness losses (Table 3). On the other hand, LSSm provides the smaller average reduction (14.6% - varying between 0.05% and 24.0%). However, the time spent to produce these small reductions is considerable, leading this method to have one of the worst overall speedups (Table 7).

## 5.2 Effectiveness of BCos

Here, we report the effectiveness results achieved by our BCos approach, comparing it with the other SOTA approaches. We emphasize that all our competitors do not make use of Quantum Computing technologies. We also consider the T-test with 95% confidence level with Bonferroni correction to understand whether there are statistically significant differences between BERT trained on the original dataset and BERT trained on the retrieved subsets according to the considered IS approaches.

As we can see in Table 3, E2SC is the only IS baseline capable of maintaining the effectiveness of the trained BERT model after the IS process in *all* datasets. LSBo, LSSm, and CNN have produced losses in at least one of the datasets. In particular, no results regarding LSBo, LSSm, and CNN could be reported in the case of the AG News dataset, our largest benchmark, as these methods have scalability problems for large datasets, as described in Section 2.3. This is consistent with the results reported in [8].

We remind that due to noise, randomness, embedding quality, and/or other external factors, the QA approach could produce higher reduction rates of the original datasets than the baselines, thus causing a slight decrease in terms of overall effectiveness with respect to the SA approach, which does not suffer from these issues since the algorithm is not performed on a quantum computer. In any case, despite such a possibility, both of our annealing solutions,

the traditional BCos(SA) and the quantum BCos(QA), are very competitive with the baselines, keeping effectiveness statistically tied in all tested datasets, being as good as the state-of-the-art method E2SC in this goal of the tripod constraints.

Finally, it is very interesting to notice that both BCos(QA) and BCos(SA) are statistically tied in terms of effectiveness showing that QA not only can help in providing a greater speedup as datasets grow in size but also in achieving comparable effectiveness.

## 5.3 Efficiency of BCos

Table 4 shows the time required to perform IS by each approach. We can see that BCos(QA) becomes more efficient as the dataset increases. In fact, as shown in Section 2.1, QA requires to perform additional steps with respect to SA such as computing the minor embedding and transferring data over the network, introducing latencies, since quantum annealers operate in the cloud. However, the effects of these parts can be mitigated in the case of large datasets, and since the actual Annealing process is much faster on quantum annealers, we can see that QA becomes more efficient than SA.

**Table 4: Time required in seconds to perform the considered SOTA IS baselines and our BCos approach, considering both SA and QA, on different datasets.**

| End-to-end time (s) required to perform the IS algorithms. | | | | | | |
|---|---|---|---|---|---|---|
| Dataset | E2SC | LSBo | LSSm | CNN | BCos (SA) | BCos (QA) |
| Vader NYT | 0.20 | 18.67 | 9.89 | 5.94 | 13.29 | 65.78 |
| Yelp Revi. | 0.61 | 16.98 | 10.36 | 13.68 | 13.68 | 39.38 |
| WebKB | 1.07 | 46.53 | 29.81 | 35.75 | 21.96 | 94.31 |
| 20 NewsG. | 3.71 | 337.07 | 192.19 | 159.76 | 51.40 | 86.25 |
| OHSUMED | 3.50 | 256.09 | 161.59 | 146.27 | 43.58 | 92.91 |
| AG News | 12.11 | - | - | - | 295.88 | 287.33 |

Table 5 shows the breakdown of the times required by BCos using both SA and QA. In particular, in that table, we consider as Annealing time the contributions of the Annealing time per core in the case of SA. It is clearly visible how the Annealing time of QA is far lower with respect to the Annealing time of its SA counterpart. We can also notice that much of the time required in the case of QA is due to latencies and minor embedding. However, the embedding time is independent of the dataset size, so it plays a minor role in the case of bigger datasets.

**Table 5: The breakdown of the time in seconds required to perform BCos on different datasets using SA and QA. QUBO Problems Formulation refers to the total time required to formulate the QUBO problem for all the batches, Embedding time is the time to calculate the minor embedding, Annealing time is the *total* amount of time required by the Annealing process (for Simulated Annealing, it is the contribution of the Annealing time due to the Annealing process performed using each core), Latency represents the enqueueing time and network latencies, and End-to-end time is the total time to run the considered approach.**

| Approach | Dataset | QUBO Problems Formulation | Embedding time | Annealing time | Latency | End-to-end time |
|---|---|---|---|---|---|---|
| | | Breakdown of BCos execution time measured in seconds using SA and QA. | | | | |
| BCos (QA) | Vader NYT 2L | 1.38 | 53.62 | 3.53 | 7.24 | 65.78 |
| | Yelp Reviews 2L | 1.42 | 28.43 | 3.54 | 5.98 | 39.38 |
| | WebKB | 3.23 | 73.63 | 5.58 | 11.86 | 94.31 |
| | OHSUMED | 4.73 | 56.80 | 13.02 | 18.33 | 92.91 |
| | 20 Newsgroups | 6.01 | 54.76 | 13.50 | 11.96 | 86.25 |
| | AG News | 31.56 | 73.17 | 87.84 | 94.60 | 287.33 |
| BCos (SA) | Vader NYT 2L | 1.47 | - | 42.79 | - | 13.29 |
| | Yelp Reviews 2L | 1.51 | - | 44.04 | - | 13.68 |
| | WebKB | 3.21 | - | 66.09 | - | 21.96 |
| | OHSUMED | 4.26 | - | 149.84 | - | 43.58 |
| | 20 Newsgroups | 6.30 | - | 165.68 | - | 51.40 |
| | AG News | 31.56 | - | 1019.16 | - | 295.88 |

**Table 6: Time required in seconds to train BERT on the different datasets and extracted subsets according to the considered SOTA IS baselines and Bcos using both SA and QA.**

| Dataset | Original Dataset | E2SC | LSBo | LSSm | CNN | BCos (SA) | BCos (QA) |
|---|---|---|---|---|---|---|---|
| | BERT Training time measured in seconds on the different datasets. | | | | | | |
| Vader NYT 2L | 204.69 | 156.56 | 119.69 | 202.22 | 136.14 | 152.18 | 143.34 |
| Yelp Reviews 2L | 234.11 | 161.88 | 76.93 | 173.07 | 123.16 | 166.18 | 165.96 |
| WebKB | 391.20 | 309.34 | 138.04 | 307.58 | 239.04 | 306.17 | 295.97 |
| 20 Newsgroups | 986.52 | 745.92 | 753.28 | 1002.16 | 711.30 | 750.51 | 723.94 |
| OHSUMED | 942.66 | 774.13 | 328.42 | 723.13 | 517.67 | 674.02 | 692.60 |
| AG News | 4675.70 | 3801.80 | - | - | - | 3773.78 | 3800.97 |
| Average Speedup | - | 1.31x | 2.35x | 1.18x | 1.65x | 1.33x | 1.35x |

**Table 7: Overall time in seconds calculated as the total time required by the IS and BERT training with corresponding speedups. It is reported the speedup considering the text representation time and without it for completeness.**

| Dataset | Original Dataset | E2SC | LSBo | LSSm | CNN | BCos (SA) | BCos (QA) |
|---|---|---|---|---|---|---|---|
| | Total time in seconds considering training BERT and applying the IS approaches. | | | | | | |
| Vader NYT 2L | 204.69 | 156.76 | 138.36 | 212.12 | 142.08 | 179.41 | 223.07 |
| Yelp Reviews 2L | 234.11 | 162.48 | 93.90 | 183.42 | 133.56 | 193.91 | 219.39 |
| WebKB | 391.20 | 310.41 | 184.57 | 307.58 | 274.79 | 347.81 | 409.96 |
| 20 Newsgroups | 986.52 | 749.63 | 1090.35 | 1194.35 | 871.06 | 840.45 | 848.73 |
| OHSUMED | 942.66 | 730.92 | 584.51 | 884.72 | 663.94 | 755.03 | 822.95 |
| AG News | 4675.70 | 3813.91 | - | - | - | 4298.72 | 4317.36 |
| **Average overall Speedup** | - | 1.31x | 1.72x | 1.06x | 1.43x | 1.16x | 1.06x |
| **Average Speedup (wo Rep. Time)** | - | 1.31x | 1.72x | 1.06x | 1.43x | 1.24x | 1.11x |

Table 6 shows the BERT training time according to the considered datasets and methods. We can see from that table that all the approaches allow to reduce the BERT training time with speedups between 1.18x-2.35x. The LSBo's good speedup is due to its very large reduction rate (56.4% on average), which brings, as a consequence, several effectiveness losses. Indeed, in terms of effectiveness, LSBo is one of the worst methods, producing losses in half of the cases (counting AG News, where it cannot be run). In this scenario, BCos(QA) achieves the 3rd best speedup among all alternatives, being able to run in all datasets, differently from LSBo

or CNN (the runner-up in terms of speedup), which also could not be run in AG News. Considering all the tradeoffs, BCos(QA) stands out as one of the best methods among all analyzed.

Finally, we can see in Table 7 the total cost, expressed as the sum of BERT training time and IS execution time, taking into account also the initial conversion of the text dataset into contextual embeddings for BCos. In almost all cases the time to train BERT after the IS selection is lower than the amount of time spent on training BERT on the original datasets. In any case, even when counting the time spent for generating the contextual embeddings, BCos(QA)

has a speedup as good as LSSm being more effective. BCos(SA) is also very competitive in this scenario.

Since creating the document representation (in any of the IS approaches) is not intrinsic to the IS task but a necessary step for any learning task in which the reduced training dataset will be used, we also consider the speed-up without taking into the time spent for building the text representations (last row of Table 7). In this scenario, the results are even better, with BCos (SA) and BCos (QA) being able to achieve speed-ups of 1.24x and 1.11x, respectively.

Overall, when combining effectiveness, training set reduction, and speedup results, we can see that BCos(SA) and BCos(QA) demonstrated a lot of potential for the IS task, especially BCos(QA), whose additional latency and embeddings costs can be reduced in the future and whose speedup can be improved due to its inherent parallelism. Moreover, our annealing formulation is simplistic and there is room for improvement to boost the effectiveness results.

## 5.4 Further Considerations

*5.4.1 Hardware Considerations.* The D-Wave Advantage quantum annealer used is a very powerful machine even though it has its own limitations due to the fact that we are still in the early stages of development of QC machines. A new quantum annealer called D-Wave Advantage 2 is expected to be released and be available soon. It will have $\approx$ 7000 qubits and a more complex topology, which allows for more connections between the qubits, thus making it possible to increase the size of the problems that can be solved. In our case, this could be useful to increase the batch size in order to boost the efficiency and effectiveness of BCos by submitting fewer batches and also considering more documents within each batch.

Indeed, preliminary studies on D-Wave Advantage 2 have shown that it could approximately *halve* the annealing time, leading to faster computations. In addition, error mitigation has been further improved, allowing for better solutions to the submitted problems [4] and improving effectiveness.

*5.4.2 Language Model Considerations.* Our current work is focused on the application of instance selection methods as a pre-processing step for methods based on 1st and 2nd generation Transformers. It should be noted that conducting the experiments presented in this paper requires a significant amount of time and resources. Given the recent rise of state-of-the-art Large Language Models (LLMs) methods, especially *open* ones such as LLama 3 [41] and Bloom [22], it is quite natural to wonder if and how the proposed instance selection methods would/could be applied to fine-tune these state-of-the-art LLMs for classification and other NLP tasks.

Indeed, we believe that the exorbitant cost of fine-tuning these LLM models – between 25-30 times more expensive than fine-tuning 1st and 2nd generation Transformers [37] – makes the application of IS methods even more appealing in these scenarios. However, these enormous costs imply that experiments with such huge models need to be more carefully planned to avoid wasting resources. Moreover it is not clear that these very complex LLMs will always be better than the best Transformer in all scenarios. For instance, RoBERTa is a remarkable sentiment classifier [5] – in some of the

datasets of the GLUE benchmark[4] RobERTa is not statistically surpassed by modern LLMs. Thus, further studies are necessary to more clearly indicate in which situations there is higher chance of obtaining (effectiveness) benefits given the huge costs of running such experiments. In other words, a deeper cost-benefit analysis is essential before we delve into this costly endeavor. This is the current focus of our research.

That said, we believe that the work presented in this paper brings enough contributions to be discussed with the scientific community. But we will, for sure, run new experiments with our Quantum IS proposals and LLMs in the foreseeable future as soon as some of the issues discussed above have clearer answers.

## 6 CONCLUSIONS AND FUTURE WORK

We have proposed a novel approach to perform IS using QA. We have shown that our approach allows to reduce the training dataset in a meaningful way, keeping the transformer's effectiveness while speeding up the model training (fine-tuning). Our approach manages to perform in line with SOTA IS approaches, but using a completely different formulation and computing paradigm, showing the potential of QC for solving real practical problems. We have also analyzed the current limitations of quantum annealers which are still to overcome. These limitations impact our practical results, but as the technology advances, we expect improved hardware capabilities and, thus, better results, both in terms of effectiveness and efficiency, allowing also to consider larger problems, datasets and language models.

In future work, we intend to experiment with new QUBO formulations, new transformers and LLMs, new datasets, new reduction rates, and new tasks besides ATC in which IS is potentially useful. Another idea is to evaluate the impact of IS solutions in bigger tasks such as the training (not the pre-training) of Large Language Models. Finally, it would be very interesting to understand the actual environmental impact of quantum annealers. In fact, reducing power and emissions is crucial and there have been attempts to analyze the emissions of several approaches in the IR field [38]. This type of analysis should also be carried out for quantum annealers to understand how much they can impact in providing greener computation.

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
