# OpenReview forum: "A Quantum Annealing Instance Selection Approach for Efficient and Effective Transformer Fine-Tuning"
_ACM.org/SIGIR/ICTIR/2024/Conference — ICTIR 2024_

### Official Review · Reviewer_eUHP · 2024-05-12

**Rating:** 1
**Confidence:** 3

**Objective Part Of Review:**

Strengths:
The submission effectively states the motivation of integrating instance selection process and quantum computing paradigms. The authors provide extensive experimental results across various benchmarks, which demonstrate the feasibility and benefits of the proposed approach.

The experiments are well-structured.

Weaknesses:
I’m not sure if the quantum annealing hardware is easily available for all researchers, especially for those in IR community. More discussion on the hardware requirements and their accessibility would be beneficial for people who are not familiar with quantum computing.

I would also appreciate a comparison with more traditional non-quantum QA baselines.

**Subjective Part Of Review:**

Most parts of the paper was easy to read and understand. The sections describing the QUBO formulation and the minor embedding process, can be challenging for readers not familiar with quantum computing. Simplifying these descriptions or providing additional explanatory materials can help. The topic is interesting but not completely related to ICTIR.

---

### Official Review · Reviewer_L6fy · 2024-05-15

**Rating:** 2
**Confidence:** 4

**Objective Part Of Review:**

The paper presents an application of quantum computing for automatic text classification, in particular for the problem of Instance Selection to reduce the size of the training set without comprimising the effectiveness of the trained classification model. Quantum Annealing, run on a D-Wave computer is used - for this a formulation of the problem that can be run on a quantum computer (QUBO fomulation) is proposed. Extensive experiments, including a comparison with simulated annealing, demonstrates how the proposed approach compared to state-of-the-art approaches. The experiments consider efficiency, effectiveness and scalability.

This application for reducing the size of the training data is gaining in importance as LLMs get larger and require more training data, but the additional training is bad for the environment.

Minor comments:
- There is a reference to "Section 3" at the beginning of Section 3.2
- It is unclear what the values in parentheses are in Table 3

**Subjective Part Of Review:**

This is an interesting paper, demonstrating a practical application of quantum computing for text classification. I learned about the considerations and formulations necessary for running algorithms on quantum computers. Hence I think that this paper will be interesting for others in the community given the increasing importance of quantum computing.

---

### Official Review · Reviewer_ZXYf · 2024-05-19

**Rating:** -1
**Confidence:** 3

**Objective Part Of Review:**

This paper proposes a novel Quantum Annealing Instance Selection approach for efficient and effective Transformer fine-tuning. The main proposal involves utilizing Quantum Computing paradigms, specifically Quantum Annealing, to address the Instance Selection (IS) problem in the context of automatic text classification (ATC) based on pre-trained language model BERT. The result in Table 6 shows the proposed approach, Balanced Cosine (Bcos), achieves the 3rd best speedup among all alternatives. Overall, the claims made in the paper are well-supported by the experimental results, analyses, and comparisons presented throughout the document, showcasing the effectiveness and competitiveness of the BCos method in the field of Instance Selection using Quantum Annealing. Some questions that arise while reading include:

1. Is the original cost function y in Eq. (2) the same as the QUBO target y in Eq. (1) ?
2. How is the penalty P(x) defined in Eq. (2)? Are they related to the Transformer Fine-Tuning models?
2. Please consider the order of the 80 examples to make Figure 2 easier to understand.
3. Explain why a batch is set to 80 instances when we have D-wave Advantage (~5000 qubits).

**Subjective Part Of Review:**

Overall, the paper is easy to follow. The authors formulate the Instance Selection (IS) problem using a Quadratic Unconstrained Binary Optimization (QUBO) approach in the context of Quantum Annealing. However, it is not clear how the QUBO problem is connected to the transformer-based fine-tuning model for ATC. Is the QUBO matrix the only connection?
Since the work is just an alternative way of instance selection, I don't find the results interesting.
I don't think others in the ICTIR community are interested in this work.

---

### Official Review · Reviewer_MAnW · 2024-05-23

**Rating:** 1
**Confidence:** 4

**Objective Part Of Review:**

In this paper, the authors present a quantum Instance Selection (IS) approach that allows to reduce the size of the training datasets (by up to 28%) while maintaining the model’s effectiveness, thus promoting (training) speedups and scalability. A series of experiments with several Automatic Text Classification (ATC) benchmarks were conducted to justify the feasibility and applicability of our proposed quantum IS approach. In their experiments, the authors adopted BERT as their Transformer-based text classification model and are going to conduct an evaluation with other Transformers such as RoBERTa, BART and XLNet in the near future.

**Subjective Part Of Review:**

After reading this paper carefully, it is still not very clear to me what the unique contribution is presented in this paper. In addition, it is somewhat outdated to discuss only the BERT-related models in 2024. It is necessary that the authors should also evaluate their proposed method with other Transformers such as RoBERTa, BART and XLNet in this paper instead of putting it as a future work.

---

### Meta-Review · Area_Chair_wVBC · 2024-06-03

**Recommendation:** Accept (Oral)
**Confidence:** 5

**Metareview:**

The paper proposes to apply Quantum Annealing, a Quantum Computing paradigm, for selecting training data for Transformer-based NLP models. The paper shows that the proposed method can reduce training data size by 28% while keeping the quality.

Strengths:
- The task for reducing the size of the training data is important
- The paper is novel
- The experiments are well-structured and convincing

Weakness:
- The paper only studies BERT but not more recent LLMs such as LLAMA, etc.
- It seems that the proposed method needs specialized quantum computing hardware. The authors should discuss how accessible these hardwares are for IR researchers.

In general this is a solid paper for ICTIR.